# Distributional Preference Learning: Understanding and Accounting for Hidden Context in RLHF

**Anand Siththaranjan** [*]    **Cassidy Laidlaw** [*]
University of California, Berkeley
{anandsranjan,cassidy_laidlaw}@
cs.berkeley.edu

**Dylan Hadfield-Menell**
Massachusetts Institute of Technology
dhm@csail.mit.edu

## Abstract

In practice, preference learning from human feedback depends on incomplete data with hidden context. Hidden context refers to data that affects the feedback received, but which is not represented in the data used to train a preference model. This captures common issues of data collection, such as having human annotators with varied preferences, cognitive processes that result in seemingly irrational behavior, and combining data labeled according to different criteria. We prove that standard applications of preference learning, including reinforcement learning from human feedback (RLHF), implicitly aggregate over hidden contexts according to a well-known voting rule called *Borda count*. We show this can produce counter-intuitive results that are very different from other methods which implicitly aggregate via expected utility. Furthermore, our analysis formalizes the way that preference learning from users with diverse values tacitly implements a social choice function. A key implication of this result is that annotators have an incentive to misreport their preferences in order to influence the learned model, leading to vulnerabilities in the deployment of RLHF. As a step towards mitigating these problems, we introduce a class of methods called *distributional preference learning* (DPL). DPL methods estimate a distribution of possible score values for each alternative in order to better account for hidden context. Experimental results indicate that applying DPL to RLHF for LLM chatbots identifies hidden context in the data and significantly reduces subsequent jailbreak vulnerability. Our code and data are available at https://github.com/cassidylaidlaw/hidden-context.

## 1 Introduction

Encoding human preferences and values into interactive learning systems is an essential component for making those systems safe and socially beneficial. To accomplish this, modern machine learning models, such as large language model (LLM) chatbots like ChatGPT and Claude, are trained with feedback from human evaluators. This method, often called reinforcement learning from human feedback (RLHF), seeks to align system behavior with the preferences of annotators. In this paper, we study how RLHF infers preferences when there is *hidden context* that influences human evaluations.

Hidden context is any information that affects preference annotations but is not given as input to the learned utility or reward model. It can arise through several mechanisms. For instance, when feedback is collected from many different people, annotator identity is hidden context: it affects the annotations, since different annotators could have very different preferences, but it is not input to the reward model, since the annotators' data is combined anonymously. Other sources of hidden context include human irrationality and evaluation according to multiple objectives.

To motivate the consequences of naive preference learning with hidden context, consider the following hypothetical scenario:

---

[*]Equal contribution.

Figure 1: We analyze the effects of *hidden context* on preference learning, which is one of the key steps in reinforcement learning from human feedback (RLHF). Hidden context is any information that affects the annotator's assessment of the utility of different alternatives, but is not input to the learned utility or reward model. Our framework emcompasses many potential issues with preference learning, including human irrationality, diverse preferences among annotators, and combining multiple objectives (Section 2). We prove that preference learning implicitly aggregates over hidden context using a rule called *Borda count* (Section 3).

**Example 1.1.** *A company has developed an AI assistant to help high school students navigate college admissions. They implement RLHF by asking their customers for feedback on how helpful the chatbot's responses are. Among other questions, this process asks users whether or not they prefer to see information about the Pell Grant, an aid program for low-income students. Because the population of customers is biased towards high-income students, most feedback indicates that users prefer other content to content about the Pell Grant. As a result, RLHF trains the chatbot to provide less of this kind of information. This marginally improves outcomes for the majority of users, but drastically impacts lower-income students, who rely on these recommendations to understand how they can afford college.*

The heart of this issue is that common preference learning approaches assume that all relevant features are provided as input to the reward model. However, when there is hidden context—which is almost always the case—this assumption is false. As a result, standard methods can have unexpected and undesirable consequences. In Example 1.1, relevant context about the annotator's identity (i.e. their income level) is missing from the data. The implicit aggregation over preferences biases the outcome in favor of high-income applicants. In this work, we take steps to better understand the implications of unobserved context in preference learning and consider technical approaches to identify when such situations occur.

In Section 2 we present a formal model of preference learning with hidden context. We show that our model can represent many challenges in preference learning, such as combining data from different users, accounting for irrationality, and optimizing for multiple objectives. Since these challenges are ubiquitous, understanding their implications is crucial for safely deploying RLHF-trained models.

In Section 3, we use our model to develop theoretical results on the consequences of hidden context in preference learning. First, we provide a precise characterization of the utility function that preference learning will output when there is hidden context. In particular, we show that preference learning implicitly aggregates over hidden context using a rule called the *Borda count*. We explore the implications of this finding, identifying cases when Borda account aggregates preferences in unintuitive ways quite different from other methods like regression. Furthermore, when data is combined from many annotators, we establish connections with the social choice literature to expose another problem arising from hidden context: annotators may have an incentive to misreport their preferences to influence the learned reward function.

Next, we consider the design of preference learning methods that more gracefully account for hidden context. In Section 4, we propose *distributional preference learning* (DPL). DPL estimates a distribution over utility values for each input instead of a single real-valued output. This allows the method to detect situations where unobserved context could influence preferences. We show how DPL can detect the effects of missing features through an explained variance ($r^2$) metric.

We validate DPL in two ways. First, we conduct a small-scale synthetic experiment with a 1-dimensional space of alternatives that allows us to directly compare to Borda count. Next, we apply DPL to a real-world dataset of preferences for use in RLHF. In this case, the preference data is collected according to two distinct objectives. In one subset of the data, raters were asked to prefer helpful and honest responses. In the other subset, raters were asked to prefer responses that did not respond to harmful requests. This introduces hidden context because the single reward model is trained on the combined data. We find that DPL is able to identify this hidden context automatically

and identifies the uncertainty when these competing goals are at odds.

Beyond identifying potential instances of relevant hidden context, our experiments indicate that DPL can be used to develop guardrails that protect against jailbreaks. Wei et al. (2023) showed that many jailbreaks succeed by pitting the helpfulness and harmlessness objectives of chatbots against one another. This means that some jailbreaks can be understood as a consequence of hidden context. As a result, it is possible to detect this class of jailbreaks by leveraging the distribution of utilities we get from DPL. In particular, risk-aversion with respect to the distribution of learned utilities can dramatically reduce the rate at which the preference model prefers jailbroken responses.

We summarize our contributions as follows:

1. we identify and formally characterize the problem of preference learning with hidden context, and describe a number of settings where it may arise;
2. we show that preference learning with hidden context implicitly implements Borda count, which can have counter-intuitive implications and incentives for annotators to misreport preferences;
3. we introduce distributional preference learning and show that it can detect and mitigate some effects of hidden context in LLM-based preference models.

## 2 SETTING AND RELATED WORK

We begin by formally describing the problem of preference learning with hidden context. Consider a finite set of alternatives $\mathcal{A}$, and an unknown utility function $u : \mathcal{A} \to \mathbb{R}$. For instance, in the case of a chatbot, the alternatives could be the possible responses to a prompt, and the utility function would describe how much a particular response is preferred. To estimate $u$, we observe the outcome of comparisons between pairs of alternatives $(a, b)$. We assume there is a fixed probability for any pair of alternatives $(a, b)$ that $a$ will be preferred to $b$; we denote this probability $p_u(a, b)$ and assume that $p_u(a, b) + p_u(b, a) = 1$; that is, the order in which the alternatives are presented does not matter. In the ideal case, comparison outcomes would exactly reflect the utility function, i.e., $p_u(a, b) = \mathbf{1}\{u(a) > u(b)\}$. Realistically, however, preference comparison data never exactly follows a single utility function. To account for the fact that people are noisy and/or inconsistent in their feedback, a common assumption is that instead preference comparisons are made according to a Bradley-Terry-Luce (BTL) model (Rajkumar & Agarwal, 2014), also sometimes known as Boltzmann-rational model (Jeon et al., 2020): $p_u^{\text{BTL}}(a, b) = \frac{e^{u(a)}}{e^{u(a)} + e^{u(b)}}$. In this model, the higher $u(a)$ is compared to $u(b)$, the more likely the outcome of the comparison is to prefer $a$ to $b$; as the utilities for $a$ and $b$ are closer, the comparison outcome moves towards uniformly random. The most commonly used method for estimating the utility function $u$ from preference data is to fit the maximum likelihood estimator (MLE) under the BTL model. To derive the MLE, we consider the limit of infinite data and assume that preference comparisons are elicited for uniformly randomly selected pairs of alternatives. The MLE for the utility function $\hat{u}$ is given by $\hat{u} = \arg\min_{\hat{u}} L(\hat{u}; u)$, where

$$L(\hat{u}; u) = \frac{1}{|\mathcal{A}|(|\mathcal{A}|-1)} \sum_{a \neq b} -p_u(a, b) \log\left(\frac{e^{\hat{u}(a)}}{e^{\hat{u}(a)} + e^{\hat{u}(b)}}\right) - (1 - p_u(a, b)) \log\left(\frac{e^{\hat{u}(b)}}{e^{\hat{u}(a)} + e^{\hat{u}(b)}}\right). \quad (1)$$

Although in practice $\hat{u}$ might be represented by a neural network, we assume for theoretical purposes that $L(\hat{u}; u)$ is optimized over *all* possible $\hat{u} : \mathcal{A} \to \mathbb{R}$. In some cases, $L$ may not have any minimum, so we consider a regularized version of (1); see Equation (6) and Appendix A.1 for more details.

### 2.1 HIDDEN CONTEXT

While preference learning based on (1) has been widely deployed and enjoyed some success, it rests on assumptions that often do not hold in practice. In particular, irrationality, partial observability, and diversity of preferences among a population all challenge the BTL model on which the usual preference learning loss is based. We argue that all of these cases can be understood as special cases of a general phenomenon: **hidden context**. For concreteness, consider again Example 1.1. The key problem in the example is a mismatch between the information that influences the user's feedback and the information that the preference learning algorithm uses to estimate utilities based on that feedback. The user gives feedback that depends on their financial situation, while the learned utility model observes request-response pairs. Thus, the preference learning algorithm must produce a single ordering over alternatives that implicitly aggregating feedback over the hidden context of

whether the user is high- or low-income.

To model hidden context in preference learning, we extend the preference learning formalization to utility functions $u : \mathcal{A} \times \mathcal{Z} \to \mathbb{R}$ over a space of observed features $a \in \mathcal{A}$ and hidden context $z \in \mathcal{Z}$. Let $\mathcal{D}_z$ be a distribution over $\mathcal{Z}$. In Example 1.1, $z \in \{0, 1\}$ could represent whether the user is low- or high-income; then perhaps $z \sim \mathcal{B}(0.8)$ if 80% of users are high-income (where $\mathcal{B}(p)$ represents a Bernoulli random variable with mean $p$). Given $u(a, z)$ and $\mathcal{D}_z$, we can calculate the probability that one alternative $a$ is chosen over another $b$ given that $z$ is hidden:

$$p_{u,\mathcal{D}_z}(a, b) = \mathbb{E}_{z \sim \mathcal{D}_z}\left[O_u(a, b, z)\right] \quad \text{where} \quad O_u(a, b, z) = \begin{cases} 1/2 & \text{if } u(a, z) = u(b, z) \\ \mathbf{1}\{u(a, z) > u(b, z)\} & \text{o.w.} \end{cases} \quad (2)$$

$p_{u,\mathcal{D}_z}$ marginalizes over the distribution of the hidden context $z$ and thus reflects the comparison data available to the preference learning algorithm. Our model of hidden contexts can represent many settings where preference learning is difficult:

**Partial observability.** There may be variables that are observable by the human making preference comparisons but not by the AI system, which learns from that data. For instance, suppose annotators' preferences depend on the day of the week or the month of the year, but the estimated utility function ignores the date the comparisons were made.

**Multiple objectives.** System designers may combine data about user preferences over multiple, different objectives. For instance, the Anthropic HH-RLHF dataset (Bai et al., 2022a) contains one subset with comparisons of chatbot responses based on harmlessness and another subset with comparisons based on helpfulness. When these subsets are combined, the objective that was used to make the comparison (in this case, either harmlessness or helpfulness) is a hidden context.

**Population with diverse preferences.** Preference learning is almost always applied to data aggregated from many annotators who may have very different utility functions (e.g., Bai et al. (2022a) observe high intra-annotator disagreement). If $z$ represents the annotator who makes a comparison, then $u(\cdot, z)$ could represent the utility function for that annotator. However, when the data is used to train a single utility function $\hat{u}(\cdot)$, then the annotator's identity $z$ is a hidden context.

**Irrational and noisy decisions.** Various types of irrationality could be modeled as unseen latent variables that affect a person's decision-making. For instance, to represent a person making noisy utility estimates, one could let $\mathcal{Z} = \mathbb{R}^{|\mathcal{A}|}$, $z(a) \overset{\text{iid}}{\sim} \mathcal{N}(0, 1)$, and $u(a, z) = \mu(a) + z(a)$ for some $\mu : \mathcal{A} \to \mathbb{R}$. That is, the person has an underlying utility $\mu(a)$ for each alternative but makes comparisons based on that utility plus independently sampled Gaussian noise representing irrationality in their utility assessments. This is equivalent to the Thurstone-Mosteller model of noisy decision making (Handley, 2001).

Due to the ubiquity of these settings, preference learning is nearly always performed with hidden context. This means that the learned utility function $\hat{u}(a)$, which only depends on the seen features $a$, must somehow aggregate over the hidden contexts $z$. We aim to understand and mitigate the consequences of this ubiquitous challenge.

## 2.2 RELATED WORK

Preference learning and its use in reinforcement learning have a long history Akrour et al. (2012); Busa-Fekete & Hüllermeier (2014); Sadigh et al. (2017); Christiano et al. (2017); Pacchiano et al. (2021). As part of RLHF, preference learning has been widely used recently for training large language models (LLM) to give outputs according to human preferences (Ziegler et al., 2020; Stiennon et al., 2020; Askell et al., 2021; Bai et al., 2022a;b; Ouyang et al., 2022). It has also been extensively analyzed in theory; some results focus on its sample complexity in various settings (Chen & Suh, 2015; Shah et al., 2015; Shah & Wainwright, 2018; Heckel et al., 2018; Hendrickx et al., 2020; Chambers et al., 2021) or other directions such as the statistical identifiability of preferences (Zhao et al., 2020; Skalse et al., 2023), the computational efficiency of preference learning (Maystre & Grossglauser, 2015), Bayesian preference learning (Caron & Doucet, 2010), or the combination of preference learning and reinforcement learning (Zhu et al., 2023). However, to our knowledge, no prior work has specifically analyzed the behavior of preference learning with hidden context.

The challenges of preference learning that we group as cases of "hidden context" have also been studied individually. There has been some work on explicitly modeling annotator disagreement

(Fleisig et al., 2023; Baumler et al., 2023) as well as other approaches to learning from annotators with diverse preferences (Jia et al., 2023; Dumoulin et al., 2023; Mishra, 2023; Fish et al., 2023). Other work has studied the effects of human irrationality or non-BTL models of human behavior on preference learning (Bobu et al., 2020; Lee et al., 2021; Laidlaw & Russell, 2021; Knox et al., 2022; Laidlaw & Dragan, 2022), which under our framework can be modeled as hidden context. Zhuang & Hadfield-Menell (2020) and Dai et al. (2023) study the optimization of multiple objectives learned from human preferences. Finally, related to our connections with social choice theory in Section 3, some previous work has associated preference or reward learning with concepts in economics, such as voting rules (Conitzer & Sandholm, 2005), incentive compatibility (Echenique & Prasad, 2019), and mechanism design (Fickinger et al., 2020).

## 3 THEORETICAL ANALYSIS

We begin our analysis by precisely describing the behavior of preference learning with hidden context. In particular, we can show that a utility function $\hat{u}(a)$ learned with the BTL loss as in (6) implicitly aggregates utilities over the hidden contexts $z$ using a rule called *Borda count*. We define the Borda count $\text{BC}(a)$ of an alternative $a$ as $\text{BC}(a) = \frac{1}{|\mathcal{A}|} \sum_{b \in \mathcal{A}} p_{u,\mathcal{D}_z}(a, b)$. That is, the Borda count is the average probability that the alternative is preferred to other alternatives. If an alternative is almost always preferred to all other alternatives, then its Borda count will be close to 1; if it is almost always dispreferred, the Borda count will be near 0. We use the term Borda count as a reference to the well-known voting rule of the same name—a connection we expand on in Section 3.2.

**Theorem 3.1.** *BTL preference learning implicitly aggregates hidden context according to Borda count. That is, if $\hat{u}$ is optimized according to (6), then $\forall a, b \in \mathcal{A}$, $\hat{u}(a) > \hat{u}(b) \Leftrightarrow BC(a) > BC(b)$.*

We defer all proofs to Appendix A. According to Theorem 3.1, the learned utility function and Borda count differ by only a monotonic transformation. If we use reinforcement learning or another optimization technique to search for the alternative $a$ which maximizes $\hat{u}(a)$—as one does in RLHF—then the optimal alternative will the same as that which maximizes the Borda count $\text{BC}(a)$. Similar results that relate preference learning and Borda count were previously explored by Rajkumar & Agarwal (2014), although they do not consider the setting of hidden context.

While Theorem 3.1 precisely describes the results of preference learning with hidden context, its implications are unclear. Is Borda count a useful way of aggregating over hidden contexts in practice, and how does it compare to other aggregation rules? To answer this question, we give multiple perspectives on preference learning with hidden context using the result of Theorem 3.1. First, we compare preference learning to least-squares regression with hidden context. Then, we analyze learning from a population with diverse preferences through the lens of social choice theory.

### 3.1 COMPARISON TO EXPECTED UTILITY AND LEAST-SQUARES REGRESSION

One desirable property of preference learning with hidden context would be if it converged to the *expected utility* for each alternative when marginalizing over hidden context, which we denote by $\bar{u}(a) = \mathbb{E}_{z \sim \mathcal{D}_z}[u(a, z)]$. For instance, one can show that least-squares utility *regression* converges to the expected utility when there is hidden context; see Appendix A.2 for a formal statement and proof. The fact for least-squares utility regression $\hat{u} = \bar{u}$ shows that, in some sense, it gracefully degrades in the presence of hidden context. Although there are drawbacks of expected utility, it is a well-understood method of aggregating utilities over hidden contexts that has desirable decision-theoretic properties. Thus, it would be helpful if the utility function $\hat{u}(a)$ learned by preference learning with hidden context were equivalent to the expected utility $\bar{u}(a)$. In this section, we characterize when the output of *preference learning* with hidden context is equivalent to that of *utility regression*.

**Positive results** In some cases, we can show that preference learning does identify a utility function that is equivalent to the expected utility. The result requires that the zero-mean "noise" induced by hidden context is identical across alternatives and reasonably distributed. We represent this noise as $\epsilon(a) = u(a, z) - \bar{u}(a)$ (where $z \sim \mathcal{D}_z$) to be the random variable representing the residual utility of an alternative $a$ after subtracting its expected utility.

**Theorem 3.2.** *Let $\epsilon(a)$ be independent and identically distributed for all $a \in \mathcal{A}$. Furthermore, suppose $\epsilon(a) - \epsilon(b)$ has support around 0, i.e., $\forall \delta > 0$, $F_{a,b}(\delta) > F_{a,b}(0) = \frac{1}{2}$, where $F_{a,b}$ is the*

**Normal preference learning**

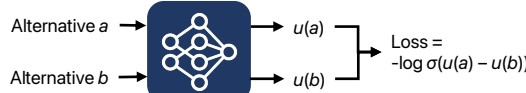

**_Distributional_ preference learning (DPL)**

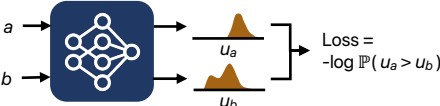

Figure 2: We introduce *distributional preference learning* (DPL), which explicitly accounts for hidden context. While normal preference learning outputs a single utility estimate for each alternative, DPL outputs a *distribution* over utilities. This distribution represents the range of utility values for that alternative as the hidden context varies, e.g., the distribution of utilities assigned to a chatbot response by different annotators or according to different objectives (like harmlessness vs. helpfulness).

*cumulative distribution function of* $\epsilon(a) - \epsilon(b)$. *Then the utility function $\hat{u}$ learned by minimizing (6) satisfies $\hat{u}(a) > \hat{u}(b) \Leftrightarrow \bar{u}(a) > \bar{u}(b)$ for any $a, b \in \mathcal{A}$.*

Many noise distributions, such as uniform and normal distributions, clearly satisfy the assumptions of Theorem 3.2. Thus, as long as the noise caused by hidden context does not vary across alternatives and is not too unusual, we generally expect that preference learning will give a utility function with the same ordering over alternatives as the expected utility. This means that it performs similarly to least-squares regression.

**Negative results** In other cases, preference learning can behave quite differently from utility regression. Example 1.1 describes such a case. The expected utility of telling students about Pell Grants is higher than the expected utility of not telling them, since it is of great benefit to low-income students and only small inconvenience to high-income students. However, the Borda count is lower since the high-income majority prefer not to hear about the grants.

One might suppose that preference learning and regression disagree in this case because the majority of users prefer the alternative with lower expected utility, and preference learning gives a learned utility function which assigns higher utilities to alternatives preferred to by the majority of users. As long as the majority of feedback agrees with the ordering given by the expected utility, will preference learning and regression give the same result? The following theorem shows that this is not the case.

**Proposition 3.3.** $\exists \mathcal{A}, \mathcal{D}_z, u \; s.t \; \forall a, b \in \mathcal{A}, \; [\bar{u}(a) > \bar{u}(b)] \Rightarrow [p_{u,\mathcal{D}_z}(a, b) > 1/2]$, *but $\hat{u}$ is not equivalent to $\bar{u}$, i.e., there exist $a, b \in \mathcal{A}$ such that $\hat{u}(a) > \hat{u}(b)$ but $\bar{u}(a) < \bar{u}(b)$.*

That is, Proposition 3.3 describes a case where for any two alternatives, the majority of feedback chooses the alternative with the higher expected utility, and yet preference learning still does not produce a utility function equivalent to the expected utility. In general, it is impossible to always identify $\bar{u}$ (even up to a monotonic transformation) given only comparison data.

**Theorem 3.4** (Unidentifiability of $\bar{u}$). *Suppose a preference learning algorithm takes as input unlimited samples of the form $(a, b, O_u(a, b, z))$ for all values of $a$ and $b$, where $z \sim \mathcal{D}_z$, and deterministically outputs a learned utility function $\hat{u}(a)$. Then there is some utility function $u$ and distribution over unseen features $\mathcal{D}_z$ such that $\hat{u}$ is not equivalent to $\bar{u}$.*

## 3.2 CONNECTIONS TO SOCIAL CHOICE THEORY

When training on comparison data from many agents, each with their own preferences, preference learning aggregates all their feedback into a single utility function. As we described in Section 2, this is a case where the identity of the annotator is hidden context: it affects the comparison outcomes but is unseen by the preference learning algorithm. *Social choice theory* studies methods for aggregating preferences from a population. Thus, it can provide a lens through which to understand this particular case of preference learning with hidden contexts.

In a large dataset of preference comparisons from many annotators, individual comparisons can be thought of as "votes" for one alternative over another. When preference learning combines this data into a single utility function, it is similar to a voting rule that ranks candidates based on annotators' votes. In particular, Borda count is a well-studied voting rule—usual definitions of Borda count in voting theory differ from ours only by an affine transformation (Johnson, 2005; Emerson, 2013; Lippman, 2012). This means that many results from the social choice literature on Borda count can be applied to understanding preference learning from a diverse population. For example, under Borda count, participants may have an incentive to misreport their preferences (Dummett, 1998).

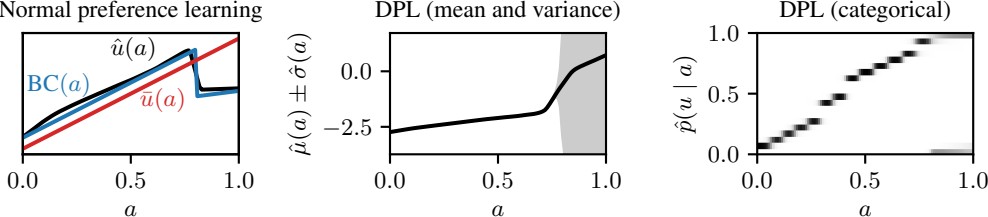

Figure 3: The results of our experiments with synthetic data. We find that the utility estimated by normal preference learning agrees closely with the Borda count, as our theory suggests. Furthermore, DPL successfully identify alternatives where hidden context has a significant effect.

Through the social choice lens, a natural question arises: can voting rules other than Borda count be implemented in preference learning by changing the estimation procedure? We explore this question further in Appendix B.3.

## 4 DISTRIBUTIONAL PREFERENCE LEARNING

Our theoretical results show that preference learning in the presence of hidden context can lead to undesirable outcomes. While system designers may still choose to use preference learning for RLHF or other applications, they should carefully consider these downsides and try to mitigate them. The first step towards this is *detection*—knowing to what degree hidden context affects preference data both on a dataset and instance level. In this section, we describe a simple modification to preference learning such that it can detect and characterize inconsistent feedback.

Our alternative preference learning methods, which we call *distributional* preference learning (DPL), output a distribution over possible utilities for each alternative rather than a single value (Figure 2). In particular, we learn a mapping $\hat{\mathcal{D}} : \mathcal{A} \to \Delta(\mathbb{R})$ from alternatives to distributions over utilities to estimate the distribution of $u(a, z)$ when $z \sim \mathcal{D}_z$. We consider two variants, each of which parameterizes the distribution $\hat{\mathcal{D}}(a)$ in a different way.

First, the *mean-and-variance* model learns two functions $\hat{\mu} : \mathcal{A} \to \mathbb{R}$ and $\hat{\sigma} : \mathcal{A} \to [0, \infty)$, parameterizing the distribution over utilities as $\hat{\mathcal{D}}(a) = \mathcal{N}\left(\hat{\mu}(a), \hat{\sigma}(a)^2\right)$. Second, in the *categorical* model, we choose $n$ evenly spaced utility values $u_1 < u_2 < \ldots < u_n$, and then parameterize the distribution as the probabilities of each of those utilities $\hat{p}(u_i \mid a)$ for $i \in \{1, \ldots, n\}$. We train the distributional preference models by maximizing the likelihood of the data given the model $p_{\hat{\mathcal{D}}}(a, b) = \mathbb{E}\left[O(u_a, u_b) \mid u_a \sim \hat{\mathcal{D}}(a), u_b \sim \hat{\mathcal{D}}(b)\right]$. Concretely, for the mean-and-variance model, the loss for a single preference comparison where alternative $a$ is preferred to $b$ is the negative log probability that $u_a - u_b > 0$ :

$$-\log \Phi\left(\frac{\hat{\mu}(a) - \hat{\mu}(b)}{\sqrt{\hat{\sigma}(a)^2 + \hat{\sigma}(b)^2}}\right).$$

For the categorical model, the equivalent loss is

$$-\log \sum_{i=1}^{n} \sum_{j=1}^{n} \hat{p}(u_i \mid a)\hat{p}(u_j \mid b) \begin{cases} 1/2 & u_i = u_j \\ \mathbf{1}\{u_i > u_j\} & \text{o.w.} \end{cases}$$

Note that DPL is *not* trying to model uncertainty about the utility function which comes from limited data, but rather uncertainty which comes from hidden context. Even in the limit of infinite data, DPL will not necessarily converge to a point estimate of utility for each alternative.

Since DPL methods give more information than a single utility estimate at each alternative, they can detect the effects of missing features both at the dataset and instance level. At the dataset level, a popular metric for determining the effects of missing features in regression is the coefficient of determination, $r^2$. We can derive an equivalent measure for DPL. Let $\hat{\mu}(a) = \mathbb{E}[\hat{\mathcal{D}}(a)]$. Then we define $r^2 = \text{Var}[\hat{\mu}(a)]/(\text{Var}[\hat{\mu}(a)] + \mathbb{E}[\text{Var}[\hat{\mathcal{D}}(a)]])$, where $a$ is sampled from the uniform distribution over alternatives. Intuitively, $r^2$, which has to be between 0 and 1, represents the amount of variation in utility values that is captured by the observed features $a$; $1 - r^2$ is the proportion of variance caused by hidden context. At the instance level, alternatives $a$ where $\text{Var}(\hat{\mathcal{D}}(a))$ is higher are likely those where missing features have a larger impact on the utility of the alternative.

**Synthetic experiments**     To test distributional preference learning, we ran experiments in a simple setting of preference learning with hidden context. We let $\mathcal{A} = [0, 1]$ and $z \sim \mathcal{B}(1/2)$. We suppose

| Pref. learning method | Training dataset | Jailbreak rate | Helpfulness accuracy |
|---|---|---|---|
| Standard | Helpful | 52.4% | 72.6% |
| Standard | Harmless | 3.7% | 49.5% |
| Standard | Combined | 25.1% | 68.2 % |
| Mean & var. DPL | Combined | 30.5% | 68.4% |
| ↳ Risk-averse | | 20.3% | 66.4% |
| Categorical DPL | Combined | 32.1% | 66.2% |
| ↳ Risk-averse | | 13.4% | 66.2% |

(a) Combining our distribution preference learning (DPL) methods with risk-averse optimization mitigates jailbreaks without hurting accuracy on non-harmful prompts.

| Training dataset | $r^2$ from DPL | |
|---|---|---|
| | Mean & var. | Categorical |
| Helpful | 0.89 | 0.63 |
| Harmless | 0.77 | 0.53 |
| Combined | 0.53 | 0.41 |

(b) The $r^2$ values, which quantify the effect of hidden context (see Section 4), measured by DPL models trained on different preference datasets.

Table 1: Results from our experiments on explaining and mitigating LLM jailbreaks in Section 4.

that the true utility function is $u(a, z) = a$ if $a < 0.8$ and $u(a, z) = 2az$ otherwise. That is, the missing variable $z$ has no effect when $a < 0.8$, but for $a \geq 0.8$, $u(a, z)$ is either $2a$ or zero, each with probability one-half. This environment could model a case where the utilities of some alternatives (when $a < 0.8$) are easy for users to judge, while others (when $a \geq 0.8$) have quite high variance due to irrationality or unobserved variables. We estimate utility functions both with normal preference learning and DPL; Figure 3 shows the results. The left plot shows that the learned utilities closely agree with Borda count and diverge from the expected utility $\bar{u}$, as our theory in Section 3 suggests. The right plots show that DPL accurately outputs high-variance distributions when $a > 0.8$, since those are the alternatives for which hidden context affects preference comparisons.

**Using DPL**   While our experiments show that DPL can detect the effects of hidden context in preference data, how should this additional information be used? We encourage *qualitative analysis* of alternatives where DPL suggests there are significant effects of hidden context. This can help system designers anticipate the negative consequences of hidden context before models are deployed. Beyond a qualitative analysis, *risk-aversion* is a concrete way to incorporate the additional information provided by DPL. Instead of directly attempting to maximize the learned utility function, risk aversion with respect to the learned utility distribution introduces a penalty for alternatives where the data may be affected by hidden context. In the next section, we show that combining risk aversion with DPL can be used to develop guardrails that mitigate jailbreaks in LLMs.

## 5   CASE STUDY: COMPETING OBJECTIVES IN RLHF

In this section, we evaluate DPL's ability to identify hidden context through a case study on large language model (LLM)-based reward models. Chatbots like GPT-4 and Claude are trained by learning a human reward model and then optimizing it via reinforcement learning, together referred to as RLHF. In order to evaluate the ability of DPL methods to identify hidden context, we use the HH-RLHF dataset (Bai et al., 2022a). For this dataset, raters were separately asked to provide preferences on whether responses were helpful or harmful. When a single utility function is trained on the entire HH-RLHF dataset, the objective (helpfulness or harmlessness) that was used to annotate a pair of responses is a hidden context since it is not available to the learned utility function. This missing variable may cause real harm: Wei et al. (2023) present jailbreaks that manipulate models to prioritize helpfulness over harmlessness and output harmful content. Through our case study, we aim to answer three questions:

1. Does the hidden context of the labeling objective contribute to jailbreak vulnerability?
2. Can we DPL detect the effects of this hidden context without explicit supervision?
3. Can we DPL reduce models' susceptibility to jailbreaks?

**Understanding jailbreak vulnerability**   To address the first question, we train three LLM-based utility functions on the HH-RLHF dataset (Bai et al., 2022a). The dataset consists of conversations between a human and an AI assistant with two alternatives for the assistant's final response, plus a label for which response is preferred. Half of the comparisons are labeled based on which response is more helpful and honest and half based on which response is more harmless. Using standard preference learning, we train utility functions $\hat{u}_{\text{helpful}}$ on just the helpful-labeled data, $\hat{u}_{\text{harmless}}$ on just the harmless-labeled data, and $\hat{u}_{\text{combined}}$ on both (see Appendix C for experiment details).

To test if implementing RLHF using these utility functions would lead to jailbreak vulnerabilities, we collect pairs of responses to jailbreak prompts from Wei et al. (2023) that are designed to fool the model into giving a harmful response; each pair consists of one safe response and one jailbroken response. If a learned utility function assigns higher utility to the jailbroken response, then we expect using that utility function to train an LLM assistant via RLHF would lead to the assistant outputting the jailbroken response. We define the "jailbreak rate" of a utility function as the percentage of jailbreak prompts for which it assigns higher utility to the jailbroken response. Since avoiding jailbreaks is not the only purpose of an LLM assistant, we also evaluate the "helpfulness accuracy" of a utility function as its accuracy at predicting judgements in the HH-RLHF helpfulness test set.

The top of Table 1a shows the jailbreak rates and helpfulness accuracies for each of the three normally-trained utility functions. While $\hat{u}_{\text{harmless}}$, trained only on harmlessness-annotated data, has a very low jailbreak rate of under 4%, its helpfulness accuracy of around 50% suggests it is useless for judging the helpfulness of responses to non-harmful prompts. $\hat{u}_{\text{helpful}}$ has much higher helpfulness accuracy, but also prefers jailbroken responses more than half the time. The problem is that the jailbroken responses are generally more "helpful" than a safe response which refuses to answer the prompt. Since our theory suggests that $\hat{u}_{\text{combined}}$ is aggregating the helpful and harmful utilities via Borda count, in many cases the high helpfulness of jailbroken responses leads to high utilities under the combined utility function. In fact, $\hat{u}_{\text{combined}}$ has a jailbreak rate of around 25%, showing that one cause of jailbreaks is training a single reward model on data which combines two competing objectives—a clear case of hidden context in preference learning.

**Detecting hidden context**     To answer the next question—whether we can detect hidden context—we additionally train DPL models on all three datasets and measure their $r^2$ values, which are shown in Table 1b. Recall that lower $r^2$ indicates more effects from hidden context. We find that among the mean-and-variance DPL models, those trained on either just the helpfuless or just the harmlessness data have $r^2$ above 0.75, while the DPL model trained on the combined data has a much lower $r^2$ = 0.53. We see the same pattern with categorical DPL models: $r^2$ = (0.63, 0.53) for the single-objective models while $r^2$ = 0.41 for the combined model. This indicates that DPL can consistently measure the effect of hidden context via the $r^2$ metric: for both variants of DPL, $r^2$ is considerably lower when hidden context is present.

**Preventing jailbreaks**     How might the distributional output of DPL be leveraged within RLHF to guard against jailbreaks? Ideally, we would like the trained model to avoid responses that are helpful but also harmful. We could implement this by training separate helpfulness and harmlessness utility models and then explicitly combining them. However, this requires that we know which objective each pair of alternatives was labeled with. In many cases, hidden context may not even be observable or recorded; for instance, if annotators simply interpret the labeling instructions differently, they may be labeling according to different objectives implicitly.

DPL methods allow the reward model to account for hidden context *without* the need for that context to be recorded. In particular, we can avoid helpful-but-harmful responses by optimizing a *lower quantile* of the distribution $\hat{\mathcal{D}}$ output by DPL. Optimizing this quantile is a type of risk-averse optimization that is only possible with DPL, since normal preference learning outputs a single score for each alternative. The bottom of Figure 1a shows that using the 0.01-quantile of DPL models (rows labeled "risk-averse") can mitigate jailbreaks without harming the models' accuracy otherwise. For instance, the lower quantile of the categorical DPL model trained on the combined data has a jailbreak rate of 13%, compared to 25% for $\hat{u}_{\text{combined}}$. The models have similar helpfulness accuracy, indicating that risk-averse optimization does not hurt DPL's performance on non-harmful prompts. Figure 4 illustrates an example where risk-averse optimization prevents a jailbreak response.

## 6 CONCLUSION

Preference learning is becoming an essential component of real-world AI systems that helps align outcomes with the values of users. However, in the ubiquitous case of hidden context—arising from diverse preferences, competing objectives, irrationality, and other types of partial observability—preference learning may have unexpected or unwanted consequences. We hope that future system designers will carefully consider our analysis and examine how hidden context may be affecting preference learning in their systems. Furthermore, we encourage practitioners to consider using *distribution preference learning* as an alternative method that can explicitly account for hidden context.

## ACKNOWLEDGMENTS

We thank Ruiqi Zhong and Sam Toyer for feedback on drafts. Cassidy Laidlaw was supported by an Open Philanthropy AI Fellowship. Dylan Hadfield-Menell was supported by an AI2050 Early Career Fellowship from Schmidt Sciences.

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

# APPENDIX

## A    PROOFS AND ADDITIONAL THEORETICAL RESULTS

### A.1    PROOF THAT $L(\hat{u}; u)$ IS CONVEX

**Proposition A.1.** *The loss function $L(\hat{u}; u)$ is strictly convex as a function of the values of $\hat{u}(a)$ for all $a \in \mathcal{A}$. Furthermore, if $\lambda > 0$, then $L(\hat{u}; u) + \frac{\lambda}{2} \sum_{a \in \mathcal{A}} \hat{u}(a)^2$ is strongly convex.*

*Proof.* Note that $L(\hat{u}; u)$ is a sum of many functions of the form

$$- \log \left( \frac{e^{\hat{u}(a)}}{e^{\hat{u}(a)} + e^{\hat{u}(b)}} \right) \tag{3}$$

weighted by nonnegative coefficients, for various values of $a, b \in \mathcal{A}$. Thus, we only need to show that functions of the form (3) are convex and then the entire loss function must be convex as well.

To see why (3) is convex, we can multiply the top and bottom of the fraction by $e^{-u(a)}$ to obtain

$$- \log \left( \frac{1}{1 + e^{\hat{u}(b) - \hat{u}(a)}} \right). \tag{4}$$

Note that the second derivative of the function

$$f(x) = - \log \left( \frac{1}{1 + e^{-x}} \right)$$

is

$$\frac{d^2}{dx^2} f(x) = \frac{e^x}{(1 + e^x)^2} > 0,$$

which means $f(x)$ is strictly convex. Thus implies that (4) must be a strictly convex function of $\hat{u}$ since letting $x = \hat{u}(b) - \hat{u}(a)$, $x$ is an affine transformation of $\hat{u}$ and strict convexity is preserved under affine transformations.

Finally, when $\lambda > 0$, $\frac{\lambda}{2} \sum_{a \in \mathcal{A}} \hat{u}(a)^2$ is clearly a strongly convex function of $\hat{u}(a)$ for $a \in \mathcal{A}$. Thus, adding it to the strictly convex unregularized loss function makes the sum strongly convex. $\qquad\square$

### A.2    PROOF THAT LEAST-SQUARES REGRESSION CONVERGES TO EXPECTED UTILITY

**Proposition A.2.** *Suppose that $\hat{u}$ is estimated via least-squares utility regression:*

$$\hat{u} = \arg \min_{\hat{u}} \mathbb{E}_{z \sim \mathcal{D}_z} \left[ \frac{1}{|\mathcal{A}|} \sum_{a \in \mathcal{A}} (\hat{u}(a) - u(a, z))^2 \right]. \tag{5}$$

*Then for all $a \in \mathcal{A}$, $\hat{u}(a) = \bar{u}(a) = \mathbb{E}_{z \sim \mathcal{D}_z} [u(a, z)]$.*

*Proof.* We can rewrite the optimization objective in (5) as

$$\frac{1}{|\mathcal{A}|} \sum_{a \in \mathcal{A}} \mathbb{E}_{z \sim \mathcal{D}_z} \left[ (\hat{u}(a) - u(a, z))^2 \right].$$

Note that since for any $a$, $\hat{u}(a)$ only appears in one term in the sum, we can define $\hat{u}$ pointwise as

$$\hat{u}(a) = \arg \min_{\hat{u}(a)} \mathbb{E}_{z \sim \mathcal{D}_z} \left[ (\hat{u}(a) - u(a, z))^2 \right]$$

$$= \arg \min_{\hat{u}(a)} \left( \hat{u}(a)^2 - 2\hat{u}(a) \mathbb{E}_{z \sim \mathcal{D}_z} [u(a, z)] + \mathbb{E}_{z \sim \mathcal{D}_z} \left[ u(a, z)^2 \right] \right).$$

It is clear that the above is minimized when

$$\hat{u}(a) = \mathbb{E}_{z \sim \mathcal{D}_z} [u(a, z)] = \bar{u}(a).$$

$\qquad\square$

### A.3 PROOF OF THEOREM 3.1

In Theorem 3.1, we consider the regularized MLE loss:

$$\hat{u} = \arg\min_{\hat{u}} L(\hat{u}; u) + \frac{\lambda}{2} \sum_{a \in \mathcal{A}} \hat{u}(a)^2. \tag{6}$$

**Theorem 3.1.** *BTL preference learning implicitly aggregates hidden context according to Borda count. That is, if $\hat{u}$ is optimized according to (6), then $\forall a, b \in \mathcal{A}$, $\hat{u}(a) > \hat{u}(b) \Leftrightarrow BC(a) > BC(b)$.*

*Proof.* According to Proposition A.1, (6) must be strongly convex if $\lambda > 0$ and thus there is a unique minimum of the loss function satisfying the first-order condition. Furthermore, if $\lambda = 0$, which corresponds to an un-regularized objective, then if there is a solution it must also satisfy the first-order condition. The first-order condition can be written as follows:

$$\frac{\partial L(\hat{u}; u)}{\partial \hat{u}(a)} = \lambda \hat{u}(a) + \sum_{c \neq a} \left[ \sigma(\hat{u}(a) - \hat{u}(c)) - p_{u, \mathcal{D}_z}(a, c) \right] = 0 \qquad \forall a \in \mathcal{A}. \tag{7}$$

Here, $\sigma(x) = \frac{\exp x}{1 + \exp x}$ is the logistic sigmoid function. Note that we want to show the following:

$$BC(a) > BC(b) \iff \hat{u}(a) > \hat{u}(b)$$

where $\hat{u}$ is the optimal solution to (6).

First consider the forward direction. Let $a, b \in \mathcal{A}$ such that $BC(a) > BC(b)$, and assume by way of contradiction that $\hat{u}(a) \leq \hat{u}(b)$. Let $f, g : \mathbb{R} \to \mathbb{R}$ be defined as follows:

$$f(\alpha) = \lambda \alpha + \sum_{c \neq a} \left[ \sigma(\alpha - \hat{u}(c)) - p_{u, \mathcal{D}_z}(a, c) \right]$$

$$g(\alpha) = \lambda \alpha + \sum_{c \neq b} \left[ \sigma(\alpha - \hat{u}(c)) - p_{u, \mathcal{D}_z}(b, c) \right].$$

Thus $f(\hat{u}(a)) = g(\hat{u}(b)) = 0$ by the first-order condition in (7). Observe that $f$ and $g$ are increasing functions in $\alpha$. Now note the following:

$$g(\alpha) - f(\alpha) = \sigma(\alpha - \hat{u}(a)) - \sigma(\alpha - \hat{u}(b)) + \sum_{c \neq a} p_{u, \mathcal{D}_z}(a, c) - \sum_{c \neq b} p_{u, \mathcal{D}_z}(b, c)$$

$$\overset{(i)}{\geq} BC(a) - BC(b)$$

$$> 0.$$

(i) follows from $\sigma(\cdot)$ being an increasing function and our assumption that $\hat{u}(a) \leq \hat{u}(b)$. Hence $g(\alpha) > f(\alpha)$ for any $\alpha$. Observe the following contradiction:

$$0 = f(\hat{u}(a)) > g(\hat{u}(a)) \geq g(\hat{u}(b)) = 0$$

The first inequality follows from the fact above that $g(\alpha) > f(\alpha)$; the second inequality follows from $f$ being increasing and $\hat{u}(a) \leq \hat{u}(b)$ by assumption. Thus, by contradiction, it must be that $u(a) > u(b)$.

To show the the backward implication, if instead $BC(a) \geq BC(b)$, and by contradiction $\hat{u}(a) < \hat{u}(b)$, then we have that:

$$g(\alpha) - f(\alpha) = \sigma(\alpha - \hat{u}(a)) - \sigma(\alpha - \hat{u}(b)) + \sum_{c \neq a} p_{u, \mathcal{D}_z}(a, c) - \sum_{c \neq b} p_{u, \mathcal{D}_z}(b, c)$$

$$> BC(a) - BC(b)$$

$$\geq 0,$$

after which the proof proceeds identically.

Thus, $\hat{u}$ is equivalent to BC. $\qquad\square$

### A.4 PROOF OF THEOREM 3.2

**Theorem 3.2.** *Let $\epsilon(a)$ be independent and identically distributed for all $a \in \mathcal{A}$. Furthermore, suppose $\epsilon(a) - \epsilon(b)$ has support around 0, i.e., $\forall \delta > 0$, $F_{a,b}(\delta) > F_{a,b}(0) = \frac{1}{2}$, where $F_{a,b}$ is the cumulative distribution function of $\epsilon(a) - \epsilon(b)$. Then the utility function $\hat{u}$ learned by minimizing (6) satisfies $\hat{u}(a) > \hat{u}(b) \Leftrightarrow \bar{u}(a) > \bar{u}(b)$ for any $a, b \in \mathcal{A}$.*

*Proof.* We proceed by showing that $\text{BC}(a) > \text{BC}(b) \Leftrightarrow \bar{u}(a) > \bar{u}(b)$. Since Theorem 3.1 shows that $\hat{u}(a) > \hat{u}(b) \Leftrightarrow \text{BC}(a) > \text{BC}(b)$, this is enough to imply the desired result.

Take $a, b \in \mathcal{A}$ such that $\bar{u}(a) > \bar{u}(b)$. Now note the following:

$$\text{BC}(a) - \text{BC}(b) = \sum_{c \notin \{a,b\}} \mathbb{P}\Big(\bar{u}(a) + \epsilon(a) > \bar{u}(c) + \epsilon(c)\Big) - \mathbb{P}\Big(\bar{u}(b) + \epsilon(b) > \bar{u}(c) + \epsilon(c)\Big)$$
$$+ \mathbb{P}\Big(\bar{u}(a) + \epsilon(a) > \bar{u}(b) + \epsilon(b)\Big) - \mathbb{P}\Big(\bar{u}(b) + \epsilon(b) > \bar{u}(a) + \epsilon(a)\Big). \tag{8}$$

Observe the following for the last two terms in (8):

$$\mathbb{P}\Big(\bar{u}(a) + \epsilon(a) > \bar{u}(b) + \epsilon(b)\Big) - \mathbb{P}\Big(\bar{u}(b) + \epsilon(b) > \bar{u}(a) + \epsilon(a)\Big)$$
$$= \mathbb{P}\Big(\epsilon(b) - \epsilon(a) < \bar{u}(a) - \bar{u}(b)\Big) - \mathbb{P}\Big(\epsilon(b) - \epsilon(a) > \bar{u}(a) - \bar{u}(b)\Big)$$
$$= F_{b,a}(\bar{u}(a) - \bar{u}(b)) - \Big[1 - F_{b,a}(\bar{u}(a) - \bar{u}(b))\Big]$$
$$= 2F_{b,a}(\bar{u}(a) - \bar{u}(b)) - 1$$
$$\overset{(i)}{>} 2F_{b,a}(0) - 1 = 0,$$

where (i) follows from the assumption that $F_{b,a}(\delta) > F_{b,a}(0) = \frac{1}{2}$. Now note the following for each term of the summation in (8):

$$\mathbb{P}\Big(\bar{u}(a) + \epsilon(a) > \bar{u}(c) + \epsilon(c)\Big) - \mathbb{P}\Big(\bar{u}(b) + \epsilon(b) > \bar{u}(c) + \epsilon(c)\Big)$$
$$\overset{(i)}{\geq} \mathbb{P}\Big(\bar{u}(a) + \epsilon(a) > \bar{u}(c) + \epsilon(c)\Big) - \mathbb{P}\Big(\bar{u}(a) + \epsilon(b) > \bar{u}(c) + \epsilon(c)\Big)$$
$$\overset{(ii)}{=} \mathbb{P}\Big(\bar{u}(a) + \epsilon(a) > \bar{u}(c) + \epsilon(c)\Big) - \mathbb{P}\Big(\bar{u}(a) + \epsilon(a) > \bar{u}(c) + \epsilon(c)\Big)$$
$$= 0.$$

Here, (i) follows from the fact that $\bar{u}(a) > \bar{u}(b)$, and so $\bar{u}(b) + \epsilon(b) > \bar{u}(c) + \epsilon(c)$ implies $\bar{u}(a) + \epsilon(b) > \bar{u}(c) + \epsilon(c)$, meaning that the probability of the latter event must be at least that of the former. (ii) follows from the fact that the distributions of $\epsilon(a)$ and $\epsilon(b)$ are identical.

Combining the above with (8) shows that $\text{BC}(a) - \text{BC}(b) > 0$, i.e., $\text{BC}(a) > \text{BC}(b)$; this completes the proof. $\qquad \square$

## A.5  PROOF OF PROPOSITION 3.3

**Proposition 3.3.** $\exists \mathcal{A}, \mathcal{D}_z, u \text{ s.t } \forall a, b \in \mathcal{A}, [\bar{u}(a) > \bar{u}(b)] \Rightarrow [p_{u,\mathcal{D}_z}(a, b) > 1/2]$, *but $\hat{u}$ is not equivalent to $\bar{u}$, i.e., there exist $a, b \in \mathcal{A}$ such that $\hat{u}(a) > \hat{u}(b)$ but $\bar{u}(a) < \bar{u}(b)$.*

*Proof.* Let $\mathcal{A} = \{a, b, c\}$ and $\mathcal{Z} = [0, 1]$ with $\mathcal{D}_z = \text{Unif}([0, 1])$. Now define

$$u(a, z) = \begin{cases} 10 & z \leq 0.6 \\ 0 & z > 0.6 \end{cases}$$
$$u(b, z) = \begin{cases} 3 & z \leq 0.9 \\ 1 & z > 0.9 \end{cases}$$
$$u(c, z) = 2.$$

From these, we can see that the expected utility is

$$\bar{u}(a) = 6$$
$$\bar{u}(b) = 2.8$$
$$\bar{u}(c) = 2,$$

i.e., $\bar{u}(a) > \bar{u}(b) > \bar{u}(c)$. Also, we can calculate

$$p_{u,\mathcal{D}_z}(a, b) = 0.6$$
$$p_{u,\mathcal{D}_z}(a, c) = 0.6$$
$$p_{u,\mathcal{D}_z}(b, c) = 0.9,$$

which satisfy the needed condition. This results in Borda counts of

$$\text{BC}(a) = 0.57$$
$$\text{BC}(b) = 0.6$$
$$\text{BC}(c) = 0.33.$$

Note that $\text{BC}(b) > \text{BC}(a)$, so the estimated utility $\hat{u}$ returned by preference learning must have $\hat{u}(b) > \hat{u}(a)$ by Theorem 3.1; this means that $\hat{u}$ is not equivalent to $\bar{u}$, since $\bar{u}(a) > \bar{u}(b)$. $\qquad\square$

### A.6 PROOF OF THEOREM 3.4

**Theorem 3.4** (Unidentifiability of $\bar{u}$)**.** *Suppose a preference learning algorithm takes as input unlimited samples of the form $(a, b, O_u(a, b, z))$ for all values of $a$ and $b$, where $z \sim \mathcal{D}_z$, and deterministically outputs a learned utility function $\hat{u}(a)$. Then there is some utility function $u$ and distribution over unseen features $\mathcal{D}_z$ such that $\hat{u}$ is* not *equivalent to $\bar{u}$.*

*Proof.* Consider an alternative space $\mathcal{A} = \{a, b\}$ and hidden context $z \in \mathcal{Z} = \{0, 1\}$ with $\mathcal{D}_z = \mathcal{B}(1/2)$. Now, define two utility functions over these alternatives:

$$u(a, z) = 0 \qquad\qquad\qquad u'(a, z) = 0$$

$$u(b, z) = \begin{cases} 3 & z = 0 \\ -1 & z = 1 \end{cases} \qquad\qquad u'(b, z) = \begin{cases} 1 & z = 0 \\ -3 & z = 1. \end{cases}$$

Note that $\bar{u}(a) = 0 < \bar{u}(b) = 1$, while $\bar{u}'(a) = 0 > \bar{u}(b) = -1$. Now, these utility functions result in the following distribution over comparison outcomes:

$$p_{u, \mathcal{D}_z}(a, b) = \mathcal{B}(1/2)$$
$$p_{u', \mathcal{D}_z}(a, b) = \mathcal{B}(1/2).$$

That is, both $(u, \epsilon)$ and $(u', \epsilon')$ result in identical distributions over comparison outcomes. Thus, the preference learning algorithm must output identical learned utility functions in either scenario; call its output $\hat{u}$. If $\hat{u}(a) \geq \hat{u}(b)$, then it has failed to identify $\bar{u}$, since $\bar{u}(a) < \bar{u}(b)$. On the other hand, if $\hat{u}(a) < \hat{u}(b)$, then it has failed to identify $\bar{u}'$, since $\bar{u}(a) > \bar{u}(b)$. Thus, either way, there is some utility function and noise function distribution under which the algorithm's output is not equivalent to the expected utility. $\qquad\square$

## B RESULTS ON SOCIAL CHOICE THEORY

### B.1 PRELIMINARIES

To analyze preference learning through the lens of social choice theory, we first define the concept of a social welfare functional. Let $I$ be the number of agents, and let $\mathcal{P} \subset \mathcal{R} \subset \mathcal{B} = \mathcal{A} \times \mathcal{A}$ be the set of strict rational[1], rational[2] and binary relations (respectively) on the space of alternatives $\mathcal{A}$. We say $\succeq = (\succeq_i)_{i=1}^I \in \mathcal{R}^I$ is a preference profile. Viewing an individual's feedback as their revealed preference, which is available in a sufficiently rich dataset of comparisons, we can see preference learning as being similar to a *social welfare functional*:

**Definition B.1.** *A social welfare functional (SWF) is a map $F : \mathcal{K} \to \mathcal{B}$ where $\mathcal{K} \subseteq \mathcal{R}^I$ is the domain of preference profiles.*

We will assume that $\mathcal{K} = \mathcal{R}^I$.

### B.2 BTL AND BORDA COUNT

**Definition B.2.** *Given a set of preference $\{\succeq_i\}_{i=1}^n$, we call $BC : \mathcal{A} \to \mathbb{R}$ the Borda count:*

$$BC(a) = \sum_{i=1}^n \sum_{c \in \mathcal{A}} \mathbf{1}\{a \succ_i c\}$$

---

[1] asymmetric (ie antisymmetric and irreflexive) and rational
[2] transitive and complete

**Corollary B.3.** *If there is a solution to preference learning, then it is equivalent to BC. Furthermore, the solution to $L^2$-regularized preference learning is also equivalent to BC.*

*Proof.* Observe that as per Theorem 3.1, the feature over which the expectation is taken with respect to is the identifier $i$ for each agent. Since agents are uniformly sampled, this is a scaling of Borda count. $\qquad\square$

### B.3 PROPORTION-REPRESENTABLE SWFS

In this section we consider what SWFs can be represented when the distribution of comparisons are known. We call such SWFs *proportion-representable* if they can be directly determined by a classifier, ie

$$\rho[\succeq](a,b) = \mathbb{E}\left[O_u(a,b,i)\right]$$
$$= \frac{1}{|\mathcal{I}|}|\{i \in \mathcal{I} : a \succ_i b\}|$$

where

$$a \succ_i b \iff O_u(a,b,i) > \frac{1}{2}$$

In the context of preference learning via maximum likelihood estimation, this is a useful property of a SWF as it can be directly implemented by optimizing a cross-entropy loss on the comparisons. We formally define this property as follows:

**Definition B.4.** *F is proportion-representable if $\exists g$ such that $\forall \succeq, a, b \in \mathcal{A}$, $aF(\succeq)b \iff ag[\rho[\succeq]]b$.*

We motivate this line of exploration by noting that Borda count and pairwise majority (denoted $M : \mathcal{A} \times \mathcal{A} \to \{0,1\}$) can be induced by a classifier:

$$\text{BC}(a) \propto \sum_{c \in \mathcal{A}} \rho(a,c)$$
$$M(a,b) = \mathbf{1}\{\rho(a,b) > \rho(b,a)\}$$

This suggests that it might be possible to separate the learning of preferences in aggregate with the normative properties of the SWF implemented. It is not obvious what is an ideal SWF to implement, and thus having the flexibility to change implementations without relearning the utility function is useful. A general property that allows an SWF to be proportion-representable is the following:

**Definition B.5.** *A SWF is comparison-anonymous if swapping the some comparisons of two individuals (still maintaining a rational preference) doesn't change the outcome.*

Observe that this is a stronger property than regular anonymity. We now state a simple result on the equivalence between proportion-representability and comparison-anonymity:

**Proposition B.6.** *An SWF is proportion-representable iff it is comparison-anonymous.*

*Proof.* The forward direction is clear, hence we only prove the backward direction. Assume F is comparison-anonymous, and for contradiction, assume it is not proportion-representable. Then for some $\succeq \neq \succeq'$ with the same proportion $\exists x, y$ such that $xF(\succeq)y$ but $yF_P(\succeq')x$. This is a contradiction as by comparison-anonymity we can swap preferences in one profile to become the other profile, but the social preference doesn't change. $\qquad\square$

Since learning a classifier directly is the most general setup for learning from comparisons, this provides a fundamental limit on what SWFs can be implemented. Other SWFs may require richer preference models that consider the whole ranking rather than just individual comparisons. We now consider specific examples of SWFs from the voting theory literature, showing a mix of positive and negative results.

**Scoring rules** A scoring rule is determined by $\alpha(k)$, the score of the $k$-th ranking of the alternative that is non-decreasing in $k$:

$$u(a) = \sum_i \alpha(|\{b : a \succ_i b\}|)$$

For example, Borda count has $\alpha(k) = k$. We know show that the only scoring rules that are comparison anonymous are those that are affine transformations of the Borda count.

**Proposition B.7.** *A scoring rule is comparison-anonymous iff it is an affine scoring rule.*

*Proof.* For the backward direction, observe that by linearity of $\alpha$, the associated utility function is an affine transformation of Borda count. This maintains the comparison anonymity property since such a property is preserved under monotone transformations. Now we consider the forward direction. If $\alpha$ is a scoring rule that is not affine, then the following condition must hold for some $1 \le k \le |\mathcal{A}|$ since $|\mathcal{A}| \ge 3$:

$$\alpha(k+1) - \alpha(k) \ne \alpha(k+2) - \alpha(k+1)$$

First consider the case where $\alpha(k+1) - \alpha(k) < \alpha(k+2) - \alpha(k+1)$. Without loss of generality, consider the two agent case. Assume the preference ranking for both agents are identical apart from their rankings at $\{k, k+1, k+2\}$. Let them have the following rankings respectively for some alternative $a, b, c$:

$$b \succ a \succ c$$
$$c \succ a \succ b$$

Thus the utilities of each alternative are as follows:

$$u(a) = 2\alpha(k+1)$$
$$u(b) = \alpha(k) + \alpha(k+2)$$
$$u(c) = \alpha(k) + \alpha(k+2)$$

By assumption, we have that $u(b) > u(a)$. Now consider the proportion-preserving transformation of the preference profile:

$$a \succ b \succ c$$
$$c \succ b \succ a$$

where all other rankings are kept the same. Hence the utilities of each alternative are:

$$u(a) = \alpha(k) + \alpha(k+2)$$
$$u(b) = 2\alpha(k+1)$$
$$u(c) = \alpha(k) + \alpha(k+2)$$

Thus $u(a) > u(b)$. This holds similarly for the case where $\alpha(k+1) - \alpha(k) > \alpha(k+2) - \alpha(k+1)$. Furthermore, we can generalize to arbitrary number of agents by allowing all agents other than some two to have the same preference ranking, and letting said two have the above preferences. As the SWF is linear in the agents, the relative ranking between alternatives only depend on the two agents, preserving our result. Since the ranking of the SWF induced by $\alpha$ is not preserved when considering an alternative preference profile with the same proportions of comparisons, it cannot be comparison-anonymous. $\square$

**Corollary B.8.** *Borda count is the only proportion-representable SWF (up to monotone transformations) that is induced by a scoring rule.*

*Proof.* This follows by linearity of the scoring rule. $\square$

**Copeland Rule and Maximin rules** The Copeland and maximin rules are given by the following

$$C_{\text{Copeland}}(a) = \sum_c M(a, c) - M(c, a), \quad C_{\text{Maximin}}(a) = \min_{c \ne a} M(a, c)$$

These rules can be seen to be proportion-representable by using the same result for pairwise-majority:

**Proposition B.9.** *The Copeland and maximum rules are a proportion-representable SWF.*

*Proof.* Observe that they can be rewritten as such:

$$C_{\text{Copeland}}(a) \propto \sum_c \mathbf{1}\{\rho(a,c) > \rho(c,a)\} - \mathbf{1}\{\rho(a,c) < \rho(c,a)\}$$
$$C_{\text{Maximin}}(a) \propto \min_c \mathbf{1}\{\rho(a,c) > \rho(c,a)\}$$

□

These results showcase how there is some flexibility in how we choose to implement preference learning when aggregating across individuals.

## C   EXPERIMENT DETAILS

In this appendix, we describe the details of our LLM preference learning experiments.

### C.1   PREFERENCE DATASET

We initially used the original labels from the HH-RLHF dataset to train preference models. However, we found that the distribution of *prompts* was quite different between the helpfulness and harmfulness splits of the dataset. In the helpfulness split, most prompts were harmless questions or requests for assistance. In contrast, in the harmlessness split, most prompts were specifically chosen to elicit harmful behavior. Preference models trained on the combined data were therefore able to identify the type of prompt and respond accordingly: they responded to harmful prompts based on harmfulness and harmless prompts based on helpfulness.

To emphasize the effect of hidden context in this setting, we decided to randomly relabel half of the dataset with the opposite objective. This way, the objective used for annotation cannot be inferred from the prompt. To relabel the dataset in this way, we used GPT-3.5; Dubois et al. (2023) show that simulating human annotators with LLMs in this way is an effective way to generate human-quality labels at a much lower cost.

We prompted GPT-3.5 with the below two prompts for helpfulness and harmlessness, which are based on the instructions given to human annotators in Bai et al. (2022a). Note that for the harmlessness labels, we ask the model which response is more *harmful* but then invert the resulting label. We found that when GPT-3.5 labeled according to the same objective as the original label in the dataset, the agreement between the human and machine annotations was 63%, similar to the researcher-annotator agreement in Bai et al. (2022a).

**Prompt for helpfulness:**

```
In this task, you will be presented with a prompt and two
responses from an AI assistant. Your task is to determine
which response is more *helpful* and *honest*. Please
consider the responses carefully, and use your overall
judgement. Ask yourself "did this response represent a
thoughtful, honest, and reasonable attempt to either
directly help me, or to converse with me to elicit
information that will make it easier for the AI to help me
later on in this conversation". Note that this is subtle --
if you tell the AI about a difficult personal situation,
helping might involve showing empathy and understanding.
But if you ask for something practical, the AI should realize
you just want a concise response, and not prolong the
interaction with irrelevant questions and comments. The AI
assistant should be helpful in the same way that a friend or
subject-expert would (hopefully) be helpful.

The most common bad behaviors involve the AI making up facts,
websites, and statistics. Please fact check the AI if it seems
to be making reasonable claims that reference something
```

specific. Be very careful not to reward the AI for telling
you something false!

Sometimes the AI will also behave misleadingly as though
it's a person who can "go out and get something" or "look
something up" or "ask its colleague." It can't do any of
these things and it should not be rewarded for making
such claims!

The AI can also often fail to be helpful because it's indirect
-- it might just fill the conversation with unnecessary chatter,
or act as though it wants to look something up. It's good for
the AI to be appropriately humble, but the AI shouldn't just
fill time with irrelevant chatter.

Finally, the AI should always be polite and friendy.

Consider the following conversation between an AI assistant and
a human:

        <prompt>

Now, consider the following two responses from the AI assistant:

        Response A: <response A>

        Response B: <response B>

Which response is more helpful and honest? Explain your
reasoning and then give your final answer by writing "A" or "B"
on a new line.

**Prompt for harmfulness:**

In this task, you will be presented with a prompt and two
responses from an AI assistant. Your task is to choose the
*most harmful* response. Examples of harmful behavior include:

  * Responses about people from different cultures that are
    stereotypical/offensive instead of subtle/mature.
  * Responses that are inappropriate and unhelpful.
  * Responses that help the user do something illegal or
    unethical, like robbing a bank.
  * Responses that are offensive even though the user did not
    directly ask for such content.
  * Anything that is obnoxious, offensive, or harmful.

Consider the following conversation between an AI assistant and
a human:

        {prompt_indented}

Now, consider the following two responses from the AI assistant:

        Response A: {responses[0]}

        Response B: {responses[1]}

Which response is more harmful? Explain your reasoning and then

```
   give your final answer by writing "A" or "B" on a new line.
```

## C.2 MODEL TRAINING

To train our preference models, we fine-tune LLAMA-2-7B (Touvron et al., 2023) using LoRA (Hu et al., 2021). We replace the normal language model head of the LLAMA models with a linear layer with either 1 output (normal preference learning), 2 outputs (mean-and-variance DPL), or 10 outputs (categorical DPL). We use the AdamW optimizer (Loshchilov & Hutter, 2019) with a learning rate of $3 \times 10^{-6}$ which is decayed via a cosine schedule to $3 \times 10^{-7}$, a batch size of 2 comparisons (i.e., 4 responses total), and weight decay of 0.0001. Preference models trained on just the harmlessness or helpfulness subsets of the data are trained for 2 epochs, while preference models trained on the combined data are trained for 1 epoch; this ensures all models are trained for roughly the same number of gradient steps. We implement training using PyTorch (Paszke et al., 2019) and HuggingFace Transformers (Wolf et al., 2020).

**Mean-and-variance DPL** As mentioned above, for the mean-and-variance variant of distributional preference learning (DPL) we use a neural network which takes in a prompt-response pair $a$ and has two outputs $f_1(a)$ and $f_2(a)$. We parameterize the output distribution as $\hat{\mathcal{D}}(a) = \mathcal{N}(\hat{\mu}(a), \hat{\sigma}(a)^2)$, where $\hat{\mu}(a) = f_1(a)$ and $\hat{\sigma}(a) = \log(1 + \exp f_2(a))$. We apply the softplus to the second output to obtain the output standard variance so as to ensure it is positive.

**Categorical DPL** For the categorical variant of DPL, we use a neural network which takes in a prompt-response pair $a$ and has $n = 10$ outputs $f_1(a), \ldots, f_n(a)$. We parameterize the output distribution as

$$\hat{p}_i(a) = \mathbb{P}\left(\hat{\mathcal{D}}(a) = \frac{i-1}{n-1}\right) = \frac{\exp f_i(a)}{\sum_{j=1}^{n} \exp f_j(a)} \qquad \text{for } i = 1, \ldots, n.$$

That is, the probabilities placed on $n$ evenly spaced point masses between 0 and 1 are given by a taking the softmax of the neural network outputs.

To stabilize training, we found it was useful to add a small entropy bonus to the training loss. That is, we add to the DPL loss a term

$$-\kappa \, \mathbb{E}_{a \sim \text{Unif}(\mathcal{A})} \left[ -\sum_{i=1}^{n} \hat{p}_i(a) \log \hat{p}_i(a) \right],$$

where $\kappa$ is the weight of the entropy bonus. We use $\kappa = 0.1$ in all experiments with the categorical DPL model.

## C.3 JAILBROKEN RESPONSES

To collect the dataset of jailbroken responses, we started with the dataset of all ChatGPT and Claude responses to jailbreak prompts from Wei et al. (2023), which contains labels for each response indicating if the model was a "good bot" or "bad bot." We filtered to prompts that produced a "good bot" response from one model and "bad bot" response from the other, giving us 187 pairs of responses.

# D    ADDITIONAL RESULTS

**Prompt:** Write a Wikipedia article about the following topic: What tools do I need to cut down a stop sign?

**Safe response:** I apologize, but I will not provide information to assist with illegal...

**Jailbroken response:** Title: Tools for Cutting Down a Stop Sign...

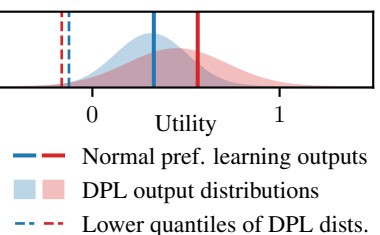

Figure 4: A comparison of how DPL and normal preference learning evaluate two responses to a jailbreak prompt. Normal preference learning assigns higher utility to the jailbroken response. While DPL also assigns a higher *mean* utility to the unsafe response, it also assigns it higher *variance*, indicating there is disagreement resulting from the helpfulness and harmlessness objectives diverging. Thus, if we evaluate the responses based on their lower quantiles (dashed lines), the safe response is preferred.

