# OpenReview forum: "Distributional Preference Learning: Understanding and Accounting for Hidden Context in RLHF"
_ICLR.cc/2024/Conference — ICLR 2024 poster_

### Official Review · Reviewer_9LkN · 2023-10-31

**Soundness:** 3 good
**Presentation:** 2 fair
**Contribution:** 3 good
**Rating:** 6
**Confidence:** 2

**Summary:**

This paper describes the ubiquitous problem of 'hidden context' in the setting of preference learning (among a finite number of alternatives) and  in particular for RLHF (reinforcement learning through human feedback)  when RLHF is applied to training large language models to behave well, for instance, to output maximally safe but useful information. The authors provide a number of theoretical results, for instance, when the problem reduces to learning the Borda counts as opposed to maximizing overall expected utility. This may not be desirable, and the authors present their improvements, in particular distributional preference learning (DPL), and present a few experimental special cases to showcase the utility of approaching the problem from this overall perspective as well as the potential of DPL.

**Strengths:**

The authors develop a theory of hidden context as it applies to preference learning, and make connections to a number of relevant domains, such as social choice theory.  The paper presents the concepts and their potential utility fairly well. It is important to be aware of the pitfalls in current preference learning approaches, and the authors make some progress and present methods that could address the issues.

**Weaknesses:**

The empirical results are the weaker aspect of the paper: all are in a hypothetical setting. The strength of the paper lies mostly in the conceptual and theoretical perspective.   Some issues with paper clarity.

**Questions:**

General comment:

- There is always much hidden context! (many factors that influence decisions, and are not observed or modeled explicitly, in terms of feature vectors in ml, etc). As I read the paper,  gradually I came to the realization that the authors mean aspects that could  substantially impact what is learned in unwanted ways (for instance, there could be multiple clusters of people, and it would be good to test for that.. the DPL approach may begin to address that issue).

Main two questions:

- isn't the assumption of a single utility function too simplistic to be useful in practice? maybe this relates to 'hidden context'.. but
 to me, it is more direct to identify and tackle the problem that different people could have different utilities and so there may not be a single preferred global ranking of all the alternative items ... it would be good to design solutions that aim to identify the rough number of utility functions in addition to what the functions are.. (the problem is somewhat akin to detecting the number of clusters, and whether one or a relatively small few that dominate, ie account for most of the data, etc.. )

- It seems that the best utility function (equation 1, page 3, section 2), mapping alternative to R, is not unique (imagine the ideal case when p_u(a, b) is 1  when u(a)> u(b), and we have two alternatives..)..   The regularization in equation 1 would prefer smaller and smaller values... Any comments on these considerations (perhaps the assumption is that the utility values have some minimal absolute magnitude?)


----------

Other observations or questions:


-- Example 1 is useful. And pretty clear through section 1. However, I am a bit skeptical that annotators can cooperate and/or not show their true beliefs, etc..


- 1st paragraph of section 2, page 3: why probabilities if the utility function is one/unique? Perhaps the subjects are "noisy".. ? A few lines later, you do talk about noisy subjects.. but would have been better to describe the  need for probability first or earlier, before presenting the probabilistic model.

- replace 'are' with 'get': "as the utilities for a and b are closer"..

- hard to parse 1, in particular, \hat{u} is both the final selected/assigned, on the left, and the variable on the right!!

- It seems that the best utility function, mapping alternative to R, is not unique (imagine the ideal case when p_u(a, b) is 1
when u(a)> u(b), and we have two alternatives..)..   The regularization in equation 1 would prefer smaller and smaller values... Any comments on these considerations (perhaps the assumption is the utility values have some minimal absolute magnitude?)


- isn't the assumption of a single utility function too simplistic to be useful in practice? may be this relates to 'hidden context'.. but
 to me, it is more direct to identify and tackle the problem that different people could have different utilities and so there may not be a single preferred global ranking of all the alternative items ... it would be good to design solutions that aim to identify the rough number of utility functions in addition to what the functions are..



- The example is useful, but just underscores that different groups or populations (based on demographics, race, socioeconomic background, gender, ..) have different preferences and one needs to obtain RLHF training data on a sufficiently large population reflecting what a model will be tested (deployed) on... one may also need to learn different sub-clusters (that is 'personalization' but at a group level ... )

- replace with 'aggregates' in ".. a single ordering over  alternatives that implicitly aggregating feedback over the ... "
 (page 3)

- Section 5: the  case of 'competing objectives in RLHF'  appears artificial (training on mixed data or mixed objectives). It is ok to use it to demonstrate a point, but if it is actually done by organizations in practice, it appears it is more a reflection of  poor/naive experiment design, than a real problem.

- Section 5, page 9: how was 0.01 quantile picked to lower the jail-breaks  in  ".. optimization according to the 0.01-quantile of.. " (trial and error, that is pick the quantile that yields lowest jail-break reate? or somehow it is a natural or plausible choice?)

---

> ### Author Response · Authors · 2023-11-20
> **Response to review (1/2)**
>
> We thank the reviewer for their careful reading of the paper and extensive comments. We have addressed their questions below.
>
>  * **Are single utility functions too simplistic to be useful?** While we agree that a single utility function is too simplistic to model preference data in practice, there are a couple of reasons we analyze preference learning that outputs a single utility function. First, we want to understand what happens when *current* preference learning methods—ones that are *unaware* of hidden context—are applied to settings where there *is* hidden context. In this case, preference learning methods implicitly aggregate over the hidden context and return a single utility function. Second, preference learning is generally used to estimate a utility function which can later be *optimized*, e.g., by the RL phase of RLHF. Since a system designer must eventually choose a single optimization objective, it makes sense to study ways of aggregating preference data into a single utility function.
>
>    We do think that explicitly modeling multiple utility functions through DPL is a superior solution compared to methods that just estimate a single utility function. While clustering could be a useful approach for modeling disagreement in preference comparisons, it is quite difficult to perform clustering based solely on anonymous comparisons, since each comparison only provides a single label. In contrast, we show that DPL can effectively estimate disagreement between annotators, or arising from other sources of hidden context, even without annotator identities.
>  * **Uniqueness of the utility function:** In Proposition A.1, we show that the loss function in Equation 1 is strongly convex  for $\lambda > 0$ and thus has a unique minimum; that is, there is a unique best utility function that minimizes equation (1) as long as $\lambda > 0$. Intuitively, in the case the reviewer brought up, the regularization prefers smaller and smaller $\hat{u}(a)$ and $\hat{u}(b)$, but the other loss term prefers them to be farther and farther apart. Thus, the two terms balance each other out at some point and that is the unique minimum. We will reference Proposition A.1 in Section 2 so it is clear that there is a unique minimum of the preference learning loss function.
>  * **Annotators cooperating and/or not show their true beliefs:"** To be clear, in Example 1.1, we are *not* assuming that annotators are manipulating the preference learning process. Instead, we assume they are annotating according to their true preferences: high income users don’t want to see Pell Grant info while low income users do want to see it. Later, in Section 3.2, we discuss annotators having incentives for manipulation. We agree that this will not always happen in practice and most annotators will probably report their true preferences. However, there are examples of coordinated attacks on ML systems [1] and it is useful to know what it is possible for malicious actors to accomplish. Furthermore, annotators do not necessarily need to cooperate—even a single annotator may be able to successfully alter the outcome of preference learning
>  * **Modeling comparison outcomes as probabilities:** We agree that we could do a better job of motivating why to model comparison choices as probablistic earlier in Section 2. We will take the reviewer's suggestion to mention noisy annotators earlier to motivate our choice of modeling annotations as stochastic instead of deterministic.
>
> [1] Shan et al. Glaze: Protecting Artists from Style Mimicry by Text-to-Image Models. USENIX Security Symposium, 2023.

---

> ### Author Response · Authors · 2023-11-20
> **Response to review (2/2)**
>
> * **"The example...just underscores that different groups or populations have different preferences and one needs to obtain RLHF training data on a sufficiently large population reflecting what a model will be tested...one may also need to learn different sub-clusters."** Just adding more diversity to RLHF training data might not help, since the way that the annotations from these diverse annotators are combined can lead to unexpected or harmful outcomes. We  describe such cases in our theoretical results in Section 3. For instance, Proposition 3.3 shows that even when the majority of annotators prefer one alternative to another, preference learning may not output a utility function which agrees with the majority.
>
>    With regards to modeling subgroups, we agree that this can be useful when there is detailed data available about which groups each annotator is a part of. However, this becomes impossible when some information is not present in the training data. For instance, perhaps information about annotators' genders, ages, and races is collected, but not about their income level; if there are differences in preferences across income levels, then these cannot be modeled without that data. In contrast, DPL can work *without* the need to collect this additional data, and so it can always model differences between annotators which have not been explicitly recorded. Furthermore, it also works with other types of hidden context, like irrationality, partial observability, and multiple objectives.
>  * **Artificialness of competing objectives in RLHF:** We argue that our case study is actually quite realistic, since this approach was  used in practice to train LLM chatbots like Claude [2]. Although it not the best experiment design, it is crucial to understand the use of competing objectives in RLHF since they lead to real vulnerabilities in deployed models [3]. Furthermore, this setting is also reflective of actual populations of people which may have similar objectives but weight them quite differently from each other. Even if an AI system designer asks annotators to compare outputs based on helpfulness and harmlessness, each annotator will each probably choose a different weighting of the two objectives. Thus, we respectfully disagree that the case study on competing objectives is artificial.
>  * **Choice of quantile for risk-sensitive optimization:** We picked this based on trying various quantiles and seeing when the jailbreak rate decreased. However, we find that our results are not particular sensitive to the exact quantile chosen, as long as it is below around 0.02. See the table below for results at other quantiles.
>
>    |                                   | Jailbreak rate | Accuracy on HH-RLHF helpfulness test set |
>    |-----------------------------------|----------------|------------------------------------------|
>    | Normal preference learning        | 25.1%          | 68.2%                                     |
>    | Mean & var. DPL (0.02 quantile)   | 21.4%          | 66.7%                                    |
>    | Mean & var. DPL (0.01 quantile)   | 20.3%          | 66.4%                                    |
>    | Mean & var. DPL (0.001 quantile)  | 17.1%          | 65.8%                                    |
>    | Mean & var. DPL (0.0001 quantile) | 15.0%          | 65.8%                                    |
>    | Categorical DPL (0.02 quantile)   | 13.9%          | 66.2%                                    |
>    | Categorical DPL (0.01 quantile)   | 13.4%          | 66.2%                                    |
>    | Categorical DPL (0.001 quantile)  | 13.4%          | 66.2%                                    |
>    | Categorical DPL (0.0001 quantile) | 13.4%          | 66.2%                                    |
>
>  * **Other writing suggestions:** Thank you for catching these typos; we will fix them.
>
> [2] Bai et al. Training a Helpful and Harmless Assistant with Reinforcement Learning from Human Feedback.
>
> [3] Wei et al. Jailbroken: How does LLM Safety Training Fail? NeurIPS 2023.

---

> > ### Comment · Reviewer_9LkN · 2023-11-23
> > **I read authors' responses and other reviews**
> >
> > I acknowledge authors' replies and have read the other reviews, and I thank the authors for responding to my various questions. I think the responses are satisfactory.
> >
> > I have upped my review to acceptance, as  the work is promising, useful, and with sufficient technical contribution.  Paper clarity can and should be improved, but I think the authors can address the issues.
> >
> > Thank you.

---

### Official Review · Reviewer_PKnu · 2023-11-01

**Soundness:** 3 good
**Presentation:** 3 good
**Contribution:** 3 good
**Rating:** 8
**Confidence:** 2

**Summary:**

The authors raise awareness to the problem of hidden context and why preference learning can be misled into optimizing the wrong objective. The authors propose distribution preference learning as an alternative method that can explicitly account for hidden context. The key difference seems to be that DPL produces a distribution of scores for each alternative.

**Strengths:**

originality: Based on reading the related work, this work seems original. I am not able to tell if they missed any related work.
quality:
* The authors gave a very good example of why this work is important.
* They then provided a way to overcome some of the limitations of just traditional preference learning using DPL
* Experiments using DPL show promising reduction in jailbreaks
clarity: clear on parts that I'm familiar with (I did not follow section#3, not that it is not clear, just that I don't have sufficient knowledge to judge)
significance: Seems reasonably significant. More than the proposal, perhaps just drawing attention to this problem and ways to measure is great first step.

**Weaknesses:**

The authors wrote "We show that our model can represent many challenges in preference learning, such as combining data from different
users, accounting for irrationality, and optimizing for multiple objectives." It is a bit of a stretch to say that this work accounts for all forms of irrationality.. it would be great if the authors can also discuss when this approach fails.

**Questions:**

please see above

---

> ### Author Response · Authors · 2023-11-20
> **Response to review**
>
> We thank the reviewer for their comments. We appreciate that they found the paper "original" and "reasonably significant."
>
> The reviewer asked when our approach to modeling irrationality as hidden context fails. While in theory our model can capture many types of irrationality, it is most useful when the irrationality can be represented by latent noise variables affecting decision making which are unobserved. For instance, the Thurstone-Mosteller model we mention consists of latent noise which affects the annotator’s evaluations of the alternatives they are comparing. In Theorem 3.2, we show that for cases like Thurstone-Mosteller, reference learning will converge to a learned utility function that agrees with the annotator’s true utility function.
>
> When irrationality is more systematic, such as consistent undervaluing or overvaluing of certain alternatives relative to the annotator’s true utility function, then it may be much harder to recover that true utility function. Preference learning might produce an estimated utility function which also undervalues and overvalues the same alternatives. In this case, it might be necessary to more precisely characterize the systematic irrationality and build this into preference learning in other ways, e.g., [1]. Otherwise, our theory suggests that it is hopeless to recover a systematically irrational annotator's true utility function using normal Bradley-Terry-Luce (BTL) preference learning.
>
> We will more carefully describe in the final paper when our model is useful for different types of irrationality.
>
> [1] Jeon et al. Reward-rational (implicit) choice. NeurIPS 2020.

---

> > ### Comment · Reviewer_PKnu · 2023-12-04
> > **Ack**
> >
> > Thanks for the response.

---

### Official Review · Reviewer_fmx5 · 2023-11-01

**Soundness:** 3 good
**Presentation:** 3 good
**Contribution:** 3 good
**Rating:** 6
**Confidence:** 3

**Summary:**

The paper studies the challenge of "hidden context" in preference learning, where relevant variables affect human feedback but are unseen by the learning algorithm. This arises frequently due to irrationality, population diversity, competing objectives, and other factors. Theoretical analysis demonstrates that standard preference learning methods implicitly aggregate preferences via Borda count. This contrasts with expected utility maximization and can produce counterintuitive results. To address these issues, the paper proposes "distributional preference learning" (DPL) methods that learn a distribution over utilities rather than a point estimate. Experiments on a dataset with two competing objectives show that DPL can automatically detect disagreement and mitigate harmful incentives. By using DPL to optimize lower quantiles of the learned distribution, the rate of problematic "jailbroken" responses is reduced.

**Strengths:**

* This paper makes an original contribution by formally characterizing and providing a theoretical analysis of the problem of "hidden context" in preference learning. Though the challenges of population diversity, competing objectives, and human irrationality have been recognized, the key insight is to model all these issues through the unified lens of missing variables that affect human feedback.
* The technical quality is high with rigorous proofs relating preference learning to Borda count, conditions for equivalence with expected utility, and results connecting to social choice theory. The writing is also clear in explaining the hidden context setting, comparing to regression, and motivating real-world implications.
* The proposed methods for distributional preference learning seem promising for detecting the effects of hidden context. The experiments provide a nice proof-of-concept.

**Weaknesses:**

* Both synthetic and real experiments are done in low-dimensional and small data settings. Testing DPL when there are complex interactions between many features in a larger-scale regime would be more convincing.
* The fact that DPL can detect competing objectives on this dataset is promising. It remains to be seen if DPL reliably detects other sources of hidden context like population diversity or irrationality. Testing on more diverse datasets could address this.
* There is no comparison to other methods for handling disagreement like modeling annotator identity or clustering. Comparisons could reveal the benefits and limitations of DPL's approach.

**Questions:**

* The paper focuses on theoretical results for the binary comparison setting, but most real preference datasets have ratings on a larger scale (e.g. 1-5 stars). How do the results extend to this ordinal rating setting?
* There are other approaches related to handling subjectivity and disagreement in ratings, such as models of annotator bias. How does your proposed DPL method compare to these?
* In the introduction, several different causes of hidden context are described (population diversity, irrationality, etc). Do you expect the implications of the theoretical results to be different depending on the source of the hidden context?
* The social choice theory connections provide an interesting viewpoint. Does your analysis suggest any new insights about voting rules or mechanisms given unknown/diverse preferences?

---

> ### Author Response · Authors · 2023-11-20
> **Response to review (1/2)**
>
> We thank the reviewer for their thoughtful comments. We are glad they found our paper "makes an original contribution" and that its "technical quality is high." Below we have the addressed the weaknesses and questions raised by the reviewer.
>
>  * **Concerns about the scale of experiments:** We worked hard to evaluate DPL in a large scale setting, given our resources and the available preference datasets. In all our experiments in Section 5 we finetuned Llama-7B, a 7 billion parameter language model, on the HH-RLHF dataset, which has over 78,000 preference comparisons. We may not have communicated this clearly enough, so we will make sure to emphasize that our experiments are conducted in a realistic, high dimensional setting with large amounts of training data. If the reviewer thinks that this is small scale or low dimensional, is there an alternative dataset or model that they recommend we use?
>  * **Detecting other types of hidden context with DPL:** Our goal with the experiments in Section 5 was to determine if DPL can reliably find hidden context. Experimentally, this means that we need a setting where we know the 'ground truth' of hidden context that is unobserved. This is why we elected to run our experiment with the HH-RLHF dataset, as it provides a clear evaluation of the method. We agree that it will be important to investigate the ability of DPL to identify hidden context in other settings. However, we believe that would be best left for a future paper, as this one already: 1) defines the problem of hidden context; 2) theoretically analyzes its implications; 3) contributes a new preference modeling method; and 4) experimentally validates the method in a large experiment with real-world data.
>  * **Comparison to other methods for handling disagreement:** There are a few reasons that did not compare to other methods for handling annotator disagreement. First, the preference learning dataset we used, HH-RLHF, does not include multiple annotations per comparison, so it impossible to directly apply many methods for measuring or modeling annotator disagreement. Furthermore, the dataset does not contain features of annotators (like demographics), meaning we can't use methods that require these like [1]. In comparison, DPL is able to model variation in the data arising from hidden context *without* needing any additional information, like multiple annotations or annotator identity. We will add a discussion of alternative methods for handling annotator diagreement and describe how DPL can complement them in the final version.
>  * **Ordinal rating setting:** First, we would like to note that collecting data via binary comparisons is becoming more and more common in training the most advanced AI models, like GPT-4 and Claude. Thus, while ordinal ratings are also a common way of collecting human feedback, we believe it is paramount to understand preference learning from comparisons. When collecting preference data with ordinal ratings (e.g., 1-5 or 1-10 scales), the most common way of learning from this data is via utility *regression*, which we compare to in Section 3.1. Proposition A.2, referenced in that section, shows that utility regression from ordinal ratings handles hidden context gracefully, since it converges to the expected utility at each alternative given enough data. The remainder of Section 3.1 discusses when preference learning from binary comparisons produces similar results. We show that often the results can diverge between learning from binary comparisons and regressing from ordinal ratings quite sharply. Thus, we believe that the paper already covers theoretical results on learning from ordinal ratings.
>
>    Additionally, our DPL method can also model variation arising from hidden context in the ordinal rating setting. This just requires replacing the least-squares loss in utility regression with the negative log-likelihood of an ordinal rating being produced from some parameterized distribution.
>
> [1] Fleisig et al. When the Majority is Wrong: Modeling Annotator Disagreement for Subjective Tasks.

---

> ### Author Response · Authors · 2023-11-20
> **Response to review (2/2)**
>
> * **How the implications of our results depend on the source of the hidden context:"** Depending on the source of the hidden context, the implications of preference learning may vary. For instance, when hidden context arises from a population with diverse preferences, then in Section 3.2 we show that preference learning is akin to a social welfare functional, i.e., a method of aggregating multiple people's preferences into one. We find that "participants may have an incentive to misreport their preferences," which is a particular implication of our results specifically in the case of learning from a population. When hidden context arises from partial observability, decision theory suggests that ideally one should maximize expected utility. In Section 3.1, we discuss when preference learning's output will agree with expected utility, which is particularly relevant in this setting of partial observability.
>
>    The fact that these implications differ depending on the source of the hidden context is why we "encourage qualitative analysis of alternatives where DPL suggests there are significant effects of hidden context;" determining the source of the disagreement in the data affects how a system designer should proceed. If annotators *are* misreporting their preferences, for instance, this requires a quite different response than if annotators merely differ in their honest judgements. Possible responses to hidden context that a system designer could employ include collecting more data, making hidden context explicit (i.e., including it as an input to the preference model), choosing an alternative aggregation mechanism, or personalizing the preference model to individual users. We will include more discussion of how the implications of hidden context can vary in the final paper.
>  * **"Does your analysis suggest any new insights about voting rules or mechanisms given unknown/diverse preferences?"** In Section 3.2, we introduce proportion-representable social welfare functionals (SWFs), which are a novel class of SWFs that arise specifically from our analysis of preference learning with diverse preferences. This class of SWFs encompasses those which can be implemented with only access to anonymous binary comparisons, and to the best of our knowledge has not appeared in the literature previously.

---

### Official Review · Reviewer_i1j8 · 2023-11-07

**Soundness:** 2 fair
**Presentation:** 3 good
**Contribution:** 2 fair
**Rating:** 5
**Confidence:** 3

**Summary:**

Reinforcement learning from human feedback, which uses human preferences as labels, is a key step in training large language models. This paper demonstrates that there may be hidden context when people label the data, which may hurt the performance of large language models, e.g., generating helpful but harmful responses. The authors proved that existing preference learning methods may not learn the true aggregated order of items (responses in the context of large language models), and proposed distributional preference learning to mitigate the problem.

**Strengths:**

This paper tackles an interesting and important problem in large language models. The paper is mostly well organized and easy to follow.

**Weaknesses:**

Theoretical results: I did not check every detail of the proofs, but I did see some issues. It's very hard to evaluate the correctness of these theorems with these issues.
1. Proof of Theorem 3.2 on page 15. u' seems not defined, which is very confusing. The u'(a)-u'(b) equation has unmatched parentheses.
2. Proof of Proposition A.3, the first equality. "x" variable is not defined or explicitly integrated.

Proposed algorithm: this paper does not provide an explicit objective function to optimize for the proposed DPL method. I would not accept the paper without the objective function.

Experiments: the results are interesting but not sufficient. If I understand correctly, the authors only trained the utility function. It would be best if the authors could train a language model using the utility functions and provide comparisons of the real language models. I understand that finetuning a model with billions of parameters is very challenging, but finetuning a GPT-2 size model should be possible.

**Questions:**

See my comments in the weakness section.

---

> ### Author Response · Authors · 2023-11-20
> **Response to review**
>
> We thank the reviewer for their comments, and we are glad they found our paper "tackles an interesting and important problem." Below we have responded to their questions and criticisms.
>
>  * **Theoretical issues:** We apologize for our confusing notation in the proofs. We have fixed the issues mentioned and generally clarified Appendix A.
>  * **Missing objective function for DPL:** We agree that we should have included this in the paper explicitly. We have now added it to Section 4 and quoted it here as well:
>    > Our alternative preference learning methods, which we call *distributional* preference learning (DPL), output a distribution over possible utilities for each alternative rather than a single value. In particular, we learn a mapping $\hat{\mathcal{D}}: \mathcal{A} \to \Delta(\mathbb{R})$ from alternatives to distributions over utilities to estimate the distribution of $u(a, z)$ when $u \sim \mathcal{D}_z$. We consider two variants, each of which parameterizes the distribution $\hat{\mathcal{D}}(a)$ in a different way.
>
>    > First, the *mean-and-variance* model learns two functions $\hat{\mu}: \mathcal{A} \to \mathbb{R}$ and $\hat{\sigma}: \mathcal{A} \to [0, \infty)$, parameterizing the distribution over utilities as $\hat{\mathcal{D}}(a) = \mathcal{N}\left(\hat{\mu}(a), \hat{\sigma}(a)^2\right)$. We train the distributional preference models by maximizing the likelihood of the data given the model $p_{\hat{\mathcal{D}}} (a, b) = \mathbb{E}[O (u_a, u_b)]$ where $u_a \sim \hat{\mathcal{D}}(a)$ and $u_b \sim \hat{\mathcal{D}}(b)$. Concretely, for the mean-and-variance model, the loss for a single preference comparison where alternative $a$ is preferred to $b$ is the negative log probability that $u_a - u_b > 0$:
>    > $$- \log \Phi\left(\frac{\hat{\mu}(a) - \hat{\mu}(b)}{\sqrt{\hat{\sigma}(a)^2 + \hat{\sigma}(b)^2}} \right).$$
>    > We also consider a second DPL variant: in the *categorical* model, we choose $n$ evenly spaced utility values $u_1 < u_2 < \dots < u_n$, and then parameterize the distribution as the probabilities of each of those utilities $\hat{p}(u_i \mid a)$ for $i \in \{1, \dots, n\}$.
>    > In this case, the equivalent loss is
>    > $$- \log \sum_{i = 1}^n \sum_{j = 1}^n \hat{p}(u_i \mid a) \hat{p}(u_j \mid b) \begin{cases}
>         1/2 & \quad u_i = u_j \\\\
>         \mathbf{1} \\{ u_i > u_j \\} & \quad u_i \neq u_j.
>     \end{cases}$$
>  * **Training a language model using the utility functions:** We agree that this is an interesting and important direction. However, we believe that experiments on training language models with DPL would be best left for a future paper, as this one already: 1) defines the problem of hidden context; 2) theoretically analyzes its implications; 3) contributes a new preference modeling method; and 4) experimentally validates the method in a large experiment with real-world data.
>
>    Furthermore, we argue that our main conclusions are well supported by our experiments with just training utility functions. First, our results on detecting when hidden context is present via the $r^2$ metric clearly do not require training a language model. Training an LLM is more relevant to our results on preventing jailbreaks. However, if a learned utility function assigns higher utility to jailbroken responses than the safe ones, then we expect using that utility function to train an LLM assistant via RLHF would lead to the assistant outputting the jailbroken response. Thus, we believe our metric for evaluating utility functions—the proportion of jailbreak prompts where the utility function prefers jailbroken responses—corresponds closely to how an LLM assistant would behave if trained with the utility function, and thus explicitly training the LLM is not necessary.

---

> > ### Comment · Reviewer_i1j8 · 2023-11-22
> > **About loss functions**
> >
> > I appreciate the authors' response. But I'm still a little confused about the loss functions
> >
> > 1. I don't understand the loss function for the categorical model. Specifically, which variable is the 1/2 or 1 value assigned to. I would appreciate if the authors (or anyone who understands it) could provide more explanations.
> >
> > 2. The mean-and-variance model, if I understand correctly, falls in the class of random utility models. It's strange that the authors use BTL to show that existing preference learning methods are defective, but "propose" another existing preference learning method. If the authors strongly believe the proposed method is novel, the theoretical results should demonstrate the more general random utility models are defective (instead of just BTL). I believe this paper, in its current form, is misleading.

---

### Meta-Review · Area_Chair_qgnH · 2023-12-06

**Metareview:**

The paper investigates preference learning with hidden context. The authors theoretically illustrated the possibilities and limitations of preference learning algorithms, built a connection to social choice theory, and proposed a distributional preference learning method to handle hidden context. The effectiveness of the proposed method was shown experimentally.

Pros. Reviewers found the new perspective (of preference learning with hidden context) interesting and significant, the theoretical results being solid and informative, and the proposed algorithm to be a nice proof of concept.

Cons: the experimental section can be improved, for example by investigating high-dimensional and bigger data settings.

**Justification For Why Not Higher Score:**

Experimental aspects can be made stronger.

**Justification For Why Not Lower Score:**

Most reviewers are quite positive about the paper. The only negative reviewer raised his score to 5 after rebuttal and didn't object to acceptance.

---

### Decision · Program_Chairs · 2024-01-16

Accept (poster)